# Deep Metric Learning for Scalable Gait-Based Person Re-Identification Using Force Platform Data

**DOI:** 10.3390/s23073392

**Published:** 2023-03-23

**Authors:** Kayne A. Duncanson, Simon Thwaites, David Booth, Gary Hanly, William S. P. Robertson, Ehsan Abbasnejad, Dominic Thewlis

**Affiliations:** 1Adelaide Medical School, The University of Adelaide, Adelaide, SA 5000, Australia; simon.thwaites@adelaide.edu.au (S.T.); david.booth@adelaide.edu.au (D.B.); dominic.thewlis@adelaide.edu.au (D.T.); 2Defence Science and Technology Group, Department of Defence, Adelaide, SA 5000, Australia; gary.hanly@defence.gov.au; 3School of Mechanical Engineering, The University of Adelaide, Adelaide, SA 5000, Australia; will.robertson@adelaide.edu.au; 4Australian Institute for Machine Learning, The University of Adelaide, Adelaide, SA 5000, Australia; ehsan.abbasnejad@adelaide.edu.au

**Keywords:** gait recognition, biometric, ground reaction force, deep learning, zero-shot re-ID, center of pressure, time series, gait analysis, force plate, classification

## Abstract

Walking gait data acquired with force platforms may be used for person re-identification (re-ID) in various authentication, surveillance, and forensics applications. Current force platform-based re-ID systems classify a fixed set of identities (IDs), which presents a problem when IDs are added or removed from the database. We formulated force platform-based re-ID as a deep metric learning (DML) task, whereby a deep neural network learns a feature representation that can be compared between inputs using a distance metric. The force platform dataset used in this study is one of the largest and the most comprehensive of its kind, containing 193 IDs with significant variations in clothing, footwear, walking speed, and time between trials. Several DML model architectures were evaluated in a challenging setting where none of the IDs were seen during training (i.e., zero-shot re-ID) and there was only one prior sample per ID to compare with each query sample. The best architecture was 85% accurate in this setting, though an analysis of changes in walking speed and footwear between measurement instances revealed that accuracy was 28% higher on same-speed, same-footwear comparisons, compared to cross-speed, cross-footwear comparisons. These results demonstrate the potential of DML algorithms for zero-shot re-ID using force platform data, and highlight challenging cases.

## 1. Introduction

Gait is a characteristic pattern of movements employed to move from one place to another [1]. Human bipedal walking gait has been proposed as a cue to identity, i.e., a biometric, that can be exploited for various authentication, surveillance, and forensics applications [2,3,4]. The attractiveness of gait as a biometric is that it can be measured unobtrusively and discreetly [3]. Based on early medical and psychological studies on the characterization of gait, it has been said to be unique within individuals, consistent over time, and difficult to obfuscate [5,6]. Therefore, walking gait may be used to control access to secure areas (e.g., within government premises, banks, and research laboratories), detect threats to national security in confined public spaces (e.g., airports and sporting venues), or identify movement of criminals retrospectively. It may be particularly useful in dynamic environments, where a face may be partially or totally occluded by other individuals, or covered (e.g., by a face mask), and other structural features are unobtainable.

Re-identification (re-ID) is the process of matching biometrics over time to re-establish identity. The re-ID task has previously been attempted using a variety of sensing modalities (e.g., video cameras, inertial sensors, pressure sensors, force platforms, and continuous wave radars) that generally acquire data on either body motion or applied forces during gait [7]. In the vast majority of systems, video cameras are utilized to acquire data pertaining to two-dimensional (2D) body appearance and motion. However, such approaches are not universally applicable, due to limitations related to body appearance (e.g., clothing, objects, and camera angle) and visibility (e.g., lighting, occlusion, and background noise) [8,9,10,11]. Thus, it is necessary to investigate the viability of complementary sensors that can provide biometric data in settings where vision-based systems are inapplicable or undependable.

Force platforms are a natural complement to video cameras because they circumvent the aforementioned limitations and are a rich source of kinetic data (i.e., data pertaining to the forces that drive motion). These devices measure ground reaction forces (GRFs) along three orthogonal axes ($F_{x}$—mediolateral, $F_{y}$—anteroposterior, and $F_{z}$—vertical) and corresponding ground reaction moments (GRMs) about each axis ($M_{x}$, $M_{y}$, and $M_{z}$). The coordinates of the GRF vector origin, referred to as the center of pressure (COP) coordinates ($C_{x}$ and $C_{y}$), are calculated from directional components of the GRFs and GRMs [12]. The directional components of each parameter result in an eight-channel time series that can be acquired during both left and right stance phases of the gait cycle if two or more force platforms are embedded into a ground surface. For reference, the gait cycle is the interval between two successive occurrences of the same gait event, such as left heel contact. The stance phase is when a given foot is in contact with the ground and comprises approximately 60% of the gait cycle. Within the stance phase, there are four functional sub-phases: loading response, mid-stance, terminal stance, and pre-swing. A detailed description of these sub-phases can be found elsewhere (e.g., [13]).

Despite the apparent prospect of force platforms as sensors for the re-ID task, there have only been a handful of studies in the field [6,14,15,16,17,18,19,20]. All of the recent works implemented machine learning classification models that were trained and evaluated at ‘many-shot’ (or subject-dependent [21]) re-ID, whereby the training set contains ‘many’ samples from the identities (IDs) that are encountered in the test set. If one of these models were to be deployed, they could only re-ID those who were included in the training set. This is a significant limitation given that IDs may be introduced or removed from the database upon deployment in many applications. In contrast to this, is the more challenging and general ‘zero-shot’ (or subject-independent [21]) re-ID, whereby the training set contains ‘zero’ samples from the IDs that are encountered in the test set (i.e., the training and test sets contain discrete IDs). Zero-shot re-ID can be achieved using deep metric learning (DML); that is, the utilization of a deep neural network (DNN) to learn a feature representation that, when compared between samples using a distance metric, is similar between samples from the same category (i.e., the same ID) and dissimilar between samples from different categories (i.e., different IDs) [22,23] (Figure 1). Using this *matching* approach, any number of IDs, not seen during training, can be re-identified at test time, provided that there is at least one prior sample. Thus, the implementation of DML would increase the scalability of force platform-based re-ID systems.

Previous studies on force platform-based re-ID also provide an incomplete picture on the effects of task and environmental constraints that are known to alter gait (Figure 2). Models have generally been trained and tested on datasets with limited or no variation in clothing, footwear, object carriage, walking speed, and time between trials [6]. The few exceptions to this have provided preliminary evidence that, on datasets with fewer than 80 IDs, these constraints can considerably reduce rank-1 accuracy (referred to herein as accuracy) using classification models (except clothing which has been assumed to have negligible effects) [16,24]. Whether or not these effects scale to larger datasets and apply to DML models remains unknown. Thus, the contributions of this work are as follows:We utilized a large force platform dataset that was purpose built for re-ID of persons and, thus, had the most complex set of walking conditions. The dataset contained 5587 walking trials from 193 IDs, with both intra- and inter-individual variations in clothing, footwear, and walking speed, as well as inter-individual variations in time between trials. A public version of the dataset, named ForceID A, contains data from 184 IDs, who consented to their data being published online (see data availability statement).To provide scope for future DML model design, we evaluated several different baseline DNN architectures at zero-shot re-ID. Two-layer fully connected neural networks (FCNNs) slightly outperformed more complex architectures over seven-fold cross validation, achieving 85% accuracy in our challenging evaluation setting, where there was only one prior sample per ID available to compare with each query sample.We analyzed the combined effects of changes in walking speed and footwear between measurement instances on re-ID performance. Accuracy across all models on same-speed, same-footwear comparisons (the easiest) was 28% higher than accuracy on cross-speed, cross-footwear comparisons (the hardest). The code repository for this study can be found at https://github.com/kayneduncanson1/ForceID-Study-1.

## 2. Related Work

### 2.1. Vision-Based re-ID

The state of vision-based systems for gait-based person re-ID has been reported extensively in the literature [21]. The standard benchmark datasets in this space—namely, CASIA-B, OU-ISIR, and OU-MVLP—contain 2D video footage acquired in controlled settings (i.e., settings with sufficient lighting and no occlusions or background noise) and under controlled walking conditions (e.g., limited or no variation in walking speed, clothing, or object carriage between measurement instances) [8,25,26]. Most of the current state-of-the-art models are DML models that rely on some variant of the triplet loss function (described in detail in Section 3.1), either in isolation or in combination with another distance metric loss function [27,28,29,30,31,32,33]. These models obtained 93–98% accuracy in the ‘Normal’ walking condition on CASIA-B (zero-shot, 50 IDs) and the subset of these models that were also implemented on OU-MVLP obtained 63–89% accuracy (zero-shot, 5154 IDs) [28,29,30,32,33]. The subset of these models that were implemented on OU-ISIR obtained 99–100% accuracy (many-shot, 4007 IDs) [27,33]. In all of the aforementioned cases, walking was conducted at preferred speed in matched clothing without any carried object. When either a carry bag or coat were introduced between measurement instances for the same 50 IDs from CASIA-B, accuracy dropped to 90–94% and 72–82%, respectively [27,28,29,30,31,32]. These findings suggest that vision-based re-ID systems perform extremely well in tightly controlled conditions, yet have limited robustness to the inclusion of walking conditions that are commonly encountered in natural settings.

The sensitivity of vision-based systems to walking conditions is illustrated by the recent work of Zhu et al. [34]. The authors acquired a video dataset (**G**ait **RE**cognition in the **W**ild (GREW) dataset) of 26,345 IDs walking in diverse public environments with variations in camera angle, lighting, background, walking speed, clothing, object carriage, footwear, and walking direction, as well as the presence of occlusions to the camera field of view. Six models (three DML models and three classification models) that were state-of-the-art at the time on standard benchmark datasets were evaluated at zero-shot re-ID on a test set of 6000 IDs from this dataset. Four out of six of the models almost completely failed (accuracy: PoseGait [35]—0%, GaitGraph [36]—1%, GEINet [37]—7%, and TS-CNN [9]—14%). The other two performed considerably better (GaitPart [30]—44% and GaitSet [38]—46%), yet still performed much worse than they did on the standard benchmark datasets (e.g., accuracy on CASIA-B: GaitPart—79–96% and GaitSet—70–95% depending on the condition). Accordingly, the authors concluded that there is still much room for improvement in the performance of vision-based systems.

### 2.2. Force Platform-Based re-ID

All studies on force platform-based re-ID have been done in the many-shot setting using classification models. Connor et al. [6] provided an overview of early studies that utilized simple machine learning models (e.g., k-Nearest Neighbour (kNN), Hidden Markov model, and Support Vector Machine). Most recent works have used multiple components of force platform data as inputs to DNN models [18,19,20]. In studies where data was acquired from fewer than 60 IDs in a single session under tightly controlled walking conditions, 88–99% accuracy was obtained using FCNN models and 99%+ accuracy was obtained using one-dimensional (1D) convolutional neural network (CNN) models [18,20]. Another study utilized an 1D convolutional long short-term memory neural network (CLSTMNN) on a dataset of 79 IDs with intra- and inter-individual variations in clothing and walking speed, as well as inter-individual variations in footwear and time between trials (3–14 days, depending on the ID) [19]. The model obtained 86% validation accuracy when trained on samples from session one and then validated on samples from session two. Instead of using DNNs, Derlatka and Borowska [39] proposed an ensemble of simple machine learning classification models. This system obtained 100% accuracy on 322 IDs, who completed a single session of walking at preferred speed in personal footwear. Even though there are numerous sources of variation between these studies, that make comparisons difficult (e.g., different datasets, input representations, model designs, training and evaluation protocols, and implementation details), they collectively highlight the potential of force platform-based re-ID.

## 3. Materials and Methods

### 3.1. Problem Formulation and Loss Function

Given a query gait sample $x_{q}$ and a set of *N* prior samples $x_{i}$ with corresponding ID labels $l_{i}$, ${\left\{\left(x_{i},l_{i}\right)\right\}}_{i=1}^{N}$, the task is to predict the query ID label $l_{q}$. Note that the set is not fixed and new samples and IDs may be added at later times. Under a DML framework, this is done by learning parameters θ for a DNN $f_{\theta }$ such that, for all *i*,
(1)$$\begin{array}{c} ∥f_{\theta }\left(x_{q}\right)-f_{\theta }\left(x_{r}\right)∥<∥f_{\theta }\left(x_{q}\right)-f_{\theta }\left(x_{i}\right)∥\;,l_{q}=l_{r}\;,l_{i}\neq l_{r}\;, \end{array}$$
where $x_{r}$ is a prior sample with the same ID label as $x_{q}$. Upon evaluation time, $l_{q}$ is assigned as the label corresponding to the sample that is most similar to $x_{q}$, which should be $l_{r}$. The θ that satisfy the conditions in Equation (Equation 1) are learnt through sequential optimization. In this study, the triplet margin loss function was used to inform optimization [40]. This function minimizes the distance between samples from the same ID and maximizes the distance between samples from different IDs in feature space. It requires a ‘triplet’, comprising a query sample, referred to as an anchor $x_{a}$, a sample from the same ID as the anchor, referred to as a positive $x_{p}$, and a sample from a different ID to the anchor, referred to as a negative $x_{n}$. The loss $\bar{L}$ for a given triplet $T=(f_{\theta }\left(x_{a}\right),f_{\theta }\left(x_{p}\right),f_{\theta }\left(x_{n}\right))$ is defined as
(2)$$\bar{L}\left(f_{\theta }\left(x_{a}\right),f_{\theta }\left(x_{p}\right),f_{\theta }\left(x_{n}\right)\right)={\left[D\left(f_{\theta }\left(x_{a}\right)-f_{\theta }\left(x_{p}\right)\right)-D\left(f_{\theta }\left(x_{a}\right)-f_{\theta }\left(x_{n}\right)\right)+\mu \right]}_{+}\;,$$
where μ is a margin enforced between $x_{p}$ and $x_{n}$ relative to $x_{a}$ and ${\left[\;\cdot \;\right]}_{+}=\max\left(\;\cdot \;,0\right)$. In this study, the DNN $f_{\theta }$ was trained on varying subsets of the dataset detailed in Section 3.2 using a number of DNN architectures detailed in Section 3.4. Training was conducted on partitions of each subset referred to as mini-batches X. The method for sampling $x_{p}$ and $x_{n}$ for a given $x_{a}$ to form *T* has been shown to influence the loss and, thus, the optimization of θ [40]. The ‘batch hard’ sampling method was chosen in this study based on the $L_{2}$ distance metric. Namely, for a given $x_{a}$, given the set *P* of candidate $x_{p}$ and the set *M* of candidate $x_{n}$ in X, the hardest $x_{p}$ (i.e., the $x_{p}$ that maximizes $L_{2}$ distance when paired with $x_{a}$) and the hardest $x_{n}$ (i.e., the $x_{n}$ that minimizes $L_{2}$ distance when paired with $x_{a}$) were selected:(3)$$\begin{array}{cc} f_{\theta }\left(x_{p}\right) & =\max_{\bar{p}\in P}{∥f_{\theta }\left(x_{a}\right)-f_{\theta }\left(x_{\bar{p}}\right)∥}_{2}^{2}\;, \end{array}$$(4)$$\begin{array}{cc} f_{\theta }\left(x_{n}\right) & =\min_{\bar{n}\in M}{∥f_{\theta }\left(x_{a}\right)-f_{\theta }\left(x_{\bar{n}}\right)∥}_{2}^{2}\;. \end{array}$$

The training loss $L_{\mathrm{train}}$ was defined as
(5)$$L_{\mathrm{train}}\left(\theta ;X\right)=\frac{1}{A}\sum_{a=1}^{A}\bar{L}\left(f_{\theta }\left(x_{a}\right),f_{\theta }\left(x_{p}\right),f_{\theta }\left(x_{n}\right)\right)\;,$$
where *A* is the total number of triplets in X. Sampling hard triplets from each mini-batch allows models to learn from challenging examples (i.e., similar samples from different IDs) without oversampling outliers [40].

### 3.2. Force Platform Dataset

#### 3.2.1. Experimental Protocol

The dataset used in this study is one of the largest force platform datasets and it has the most complex set of walking conditions, because it was purpose built for evaluating person re-ID systems. It contains 5587 walking trials (i.e., walks along the length of our laboratory in one direction) from 193 IDs (54% female; mean ± std. dev.: age 27±7 years, mass 71.86±15.93 kg, and height 1.72±0.10 m), with 19–30 trials per ID. All participants gave their informed consent for inclusion before they participated in the study. The study was conducted in accordance with the Declaration of Helsinki, and the protocol was approved by the University of Adelaide Human Research Ethics Committee (H-2018-009). Participants had no known neurological or musculoskeletal disorder that could affect their gait (this simplified the experimental protocol and facilitated the necessary first step of evaluating re-ID systems on healthy individuals). In regard to personal clothing and footwear, participants each attended two sessions separated by 3–14 days depending on the ID (there were 2777 session one trials and 2810 session two trials). At the start of each session, age, sex, mass, height, and footwear type were recorded. Footwear and clothing were also photographed for future reference. Next, participants walked along the length of the laboratory in one direction (≈ 10 m) five times at each of three self-selected speed categories: preferred, slower than preferred, and faster than preferred. Two in-ground OPT400600-HP force platforms (Advanced Mechanical Technology Inc., Phoenix, AZ, USA) in the center of the laboratory measured GRFs and GRMs during left and right footsteps. These measures, along with calculated COP coordinates, were acquired through Vicon Nexus (Vicon Motion Systems Ltd., Oxford, UK) at 2000 Hz. Of note, all trials in this dataset were complete foot contacts; that is, each foot completely contacted within the area of each force platform, as identified from video footage. In this study, each sample comprised one trial to simulate a single pass through an area of interest during locomotion. ForceID A (the public version of the dataset) contains 5327 trials from 184 participants who consented to their data being published online (see data availability statement).

#### 3.2.2. Dataset Characteristics

Figure 3 provides a visual representation of the dataset used in this study. A comparison of the sets of signals for three random IDs in the dataset provided an example of how signal shape varied both within and between IDs. The overlap between colors for certain components at certain sub-phases during stance suggests that discriminating between IDs in the dataset (e.g., IDs 108 and 125) could be quite challenging and would, thus, require the identification of subtle shape features. In fact, this dataset was prospectively designed to be challenging via the inclusion of several task and environmental constraints that increase intra-individual variability and can be expected in natural settings.

Table 1 provides an overview of key characteristics of ForceID A compared to other large-scale public force platform datasets. It should be noted that almost all studies in this field have utilized private datasets with fewer than 80 IDs [6,17,19,20,24]. Exceptions to this are Horst et al. [18], who used data from 57 IDs that is available in in their public database [41], and Derlatka and Borowska [39], who used data from 322 IDs (that may be from their public dataset [42], though it was not mentioned). This makes ForceID A the second largest dataset to be implemented for force platform-based re-ID and by far the largest dataset of repeated measures over separate days. ForceID A is also the only dataset with intra- and inter-individual variations in footwear, walking speed, and clothing (Table 1). Hence, while it can be used for a variety of purposes, its primary use case is the development of practical force platform-based re-ID systems.

Table 2 shows the distribution of footwear in the private version of the dataset. Of note, 78 participants (≈40%) wore different footwear in session two compared to session one, and 67 participants (≈35%) wore a different *type* of footwear in session two compared to session one. It was assumed that almost all participants wore different footwear to one another. One participant’s footwear was unknown in session one because their pictures were corrupted. They were categorized as wearing different footwear between sessions. In terms of walking speed, there were 1892 (≈34%) slower than preferred speed samples, 1900 (≈34%) preferred speed samples, and 1795 (≈32%) faster than preferred speed samples. Even though speed was not measured directly, there was assumed to be inter-individual variation in self-selected speeds within each category [45].

#### 3.2.3. Definition of Training, Validation, and Test Sets

In these experiments, four subsets of the dataset were utilized, each containing a different combination of walking speed and footwear conditions. Each subset was labeled with notation $D_{i,j}$. *i* corresponds to whether samples from preferred speed only (PS) or all speeds (AS) were included, while *j* corresponds to whether samples from all IDs (AF), or just those who wore the same footwear between sessions (SF), were included. The subsets were:$D_{AS,AF}$—contained samples from all speeds from all IDs (i.e., the entire dataset). This subset allowed same-speed, cross-speed, same-footwear, and cross-footwear comparisons (193 IDs, 5587 samples).$D_{AS,SF}$—contained samples from all speeds from IDs who wore the same footwear between sessions. This subset allowed same-speed, cross-speed, and same-footwear comparisons (114 IDs, 3298 samples).$D_{PS,AF}$—contained samples from preferred speed from all IDs. This subset allowed same (preferred–preferred) speed, same-footwear, and cross-footwear comparisons (193 IDs, 1900 samples).$D_{PS,SF}$—contained samples from preferred speed from IDs who wore the same footwear between sessions. This subset allowed same (preferred–preferred) speed and same-footwear comparisons (114 IDs, 1122 samples).

Each of the data subsets were distributed into training, validation, and test sets. To perform zero-shot re-ID, these sets each contained a distinct group of IDs. To cycle all IDs through validation and test sets, performance was evaluated over seven-fold cross-validation. The distribution of IDs in training, validation, and test sets was approximately 70/15/15% in the first six folds and approximately 80/10/10% in the seventh fold (which contained left-over IDs). This procedure was done once for each subset so that all models were implemented on the same training, validation, and test sets.

### 3.3. Signal Pre-Processing

As can be seen in Figure 4, the four main steps to the pre-processing method were as follows:A 50 N $F_{z}$ threshold was used to clip all signals to include only the stance phase. For GRFs and GRMs, 20 frames were retained at each end as a buffer. For COP coordinates, an additional 5% of relative length was excluded at each end to avoid inaccuracies at low force values [46].All signals were low-pass filtered (4th order bi-directional Butterworth, cut-off frequency 30 Hz) to minimize high frequency noise. This is common practice for processing time series signals of walking gait kinetics [47]. The precise time points where filtered $F_{z}$ equaled 50 N were then defined via linear interpolation.All signals were time normalized via linear interpolation to time synchronize events within the stance phase and reduce the dimensionality of inputs. A temporal resolution of 300 frames was selected as a conservatively high value, given that a prior study found minimal difference between 100 vs. 1000 frame inputs in a related gait classification task [48].Since there were different measurement scales across the eight channels, the features within each channel were standardized to zero mean and unit variance using the means and standard deviations from the training set.

### 3.4. Network Architectures

#### 3.4.1. Overview

To inform future development of DML models for force-platform based re-ID, several baseline DNN architectures of varying type and complexity (in terms of expressive capacity [49]) were implemented (Figure 5):

FCNNCNNCLSTMNNConvolutional bi-directional long short-term memory neural network (C-Bi-LSTMNN)Convolutional transformer neural network (CTNN).

This was the first time a C-Bi-LSTMNN or CTNN had been implemented for force platform-based re-ID. FCNNs, CNNs, and CLSTMNNs had been implemented previously, though comparison between these architectures is difficult due to differences in re-ID system design between studies (e.g., dataset characteristics, loss function, optimizer, and implementation details [18,19,20,50]). Even if conclusions can be drawn from previous work, they may not generalize to DML models in the zero-shot setting. Hence, in this study, each of the architectures were implemented using the same overall framework to facilitate fair comparison. Ablation experiments were also conducted to determine if input feature selection was warranted under the proposed framework. Each architecture was adapted to accommodate a single channel input and implemented with an individual component of force platform data.

#### 3.4.2. Details

To generate inputs for the FCNN, the pre-processed signals from each force platform were flattened into a feature vector and, thus, the inputs were shaped (number of samples N× number of features *F*). This architecture contained two fully connected layers separated by batch normalization (BN) and exponential linear unit (ELU) activation, respectively (referred to herein as a fully connected module) [51,52]. To generate inputs for the remaining architectures, the pre-processed signals from each force platform were concatenated lengthwise and, thus, the inputs were shaped (N× number of channels C× sequence length L(=600)). These architectures each contained four 1D convolutional layers at the start (referred to herein as a convolutional module) and a fully connected module at the end. For the convolutions, kernel size was set to three and padding was set to one, such that *L* remained the same. Initially, *C* was set to 32 for the first convolution and was then increased by a factor of two for each convolution thereafter (C=32→64→128→256). Between convolutions BN, ELU activation, and local average pooling (LAP) occurred, respectively. Kernel size and stride were set to two for the first three LAP operations, such that *L* was reduced by a factor of two each time (L=300→150→75). These hyper-parameters were set to three for the fourth LAP operation, such that *L* was reduced by a factor of three (L=75→25). The CNN just contained a convolutional module and a fully connected module. The CLSTMNN and C-Bi-LSTMNN also contained a long short-term memory (LSTM) layer which was uni-directional in the former and bi-directional in the latter (input and hidden sizes were equal). The CTNN had a transformer encoder layer (with four heads and Gaussian Error Linear Unit (GELU) activation [53]) in between the convolutional and fully connected modules.

### 3.5. Performance Evaluation

The performance metrics used in this study were accuracy and F1 score [54]. When evaluating performance in a given validation or test set, session two samples were used as queries and session one samples were used as priors, constraining the task to inter-session re-ID. The common method for generating a prediction for a given query is to assign the ID label corresponding to the prior that forms the minimum distance across *all* candidate priors. A more challenging method was implemented in this study, whereby the minimum distance was taken across a subset containing the *i*-th random prior from each ID. This simulated an initial deployment case, wherein everyone passes through an area once, and then a query ID passes through a second time and must be re-identified. With *m* as the minimum number of samples per session per ID in a given validation or test set, the process was completed *m* times, each time with a different subset of priors. This made use of the multiple samples per ID and maximized the number of predictions over which the metrics were computed. For a given query $x_{q}$ and set of priors $x_{i}$, the prediction was correct if $∥f_{\theta }\left(x_{q}\right)-f_{\theta }\left(x_{r}\right){∥}_{2}^{2}<{∥f_{\theta }\left(x_{q}\right)-f_{\theta }\left(x_{i}\right)∥}_{2}^{2}$, where $x_{r}$ is a prior with the same ID label as $x_{q}$. Accuracy was defined as the percentage of correct predictions over the total number of predictions. The F1 score was computed using binarized predictions and respective ground truth labels with macro-averaging.

### 3.6. Hyper-Parameters

Different mini-batch sizes ($N_{X}=\left\{32,64,128,256,512\right\}$) were compared to determine whether altering the number of candidate positives and negatives during training affected convergence and subsequent test performance. Mini-batches were loaded using a weighted random sampler with the weights being the probabilities of sampling from each ID. The other hyper-parameters were not tuned because the purpose of this set of experiments was to present sensible and comparable baselines, rather than a single fine-tuned model. The μ hyper-parameter for the loss function was set to 0.3, based on previous experiments on person re-ID, where performance was very similar across μ={0.1,0.2,0.5} [40]. Weights were optimized using the AMSGRAD optimizer with default hyper-parameter settings (β=(0.9,0.999), $\epsilon =1\times 10^{-8}$, weight decay =0). AMSGRAD was chosen because it has been shown to guarantee convergence by relying on long-term memory of past gradients when updating model parameters [55]. The number of epochs was set to 1000 at a learning rate of 0.0001. To prevent over-fitting the training set, training was terminated if validation loss did not improve (minimum change =0) over 20 consecutive epochs.

## 4. Results

### 4.1. Main Experiments

The results presented herein are for mini-batch size 512 as it led to the best performance (Appendix A). Accuracy and F1 score were roughly equivalent within each condition, so only accuracy is mentioned in-text. Models with the FCNN architecture were the most accurate on all four subsets (85–92%, depending on the subset in question—Table 3). The amount of variation in mean accuracy, according to architecture, was 2.3% on $D_{AS,AF}$, 2.7% on $D_{AS,SF}$, 5.8% on $D_{PS,AF}$, and 4.2% on $D_{PS,SF}$. Regardless of the architecture, accuracy was lower on subsets with all IDs compared to subsets with only those who wore the same footwear between sessions (mean difference =5.8%). In all but one case, accuracy was higher on subsets with samples from all speeds compared to subsets with samples from preferred speed only (mean difference =0.97%).

The number of correct and incorrect predictions across all models for different walking speeds and footwear comparisons were tallied to determine if there were specific comparisons that generally limited re-ID accuracy on $D_{AS,AF}$. As can be seen in Table 4, accuracy across all models ranged from 68–96%, depending on the type of comparison. Accuracy on cross-footwear comparisons was, on average, 14% lower than that on same-footwear comparisons. Accuracy on cross-speed comparisons was, on average, 8.7% lower than that on same-speed comparisons. Finally, the accuracy on preferred–preferred speed comparisons within $D_{AS,AF}$ was 6.1% higher than the accuracy of preferred–preferred speed comparisons within $D_{PS,AF}$.

The accuracy across all models was calculated on a per-ID basis, and it was found that 44% of all incorrect predictions were attributed to just 20 of the 193 IDs. In this group, the mean per-ID accuracy was 45% (range: 3.0–73%) (Appendix A). Of the IDs, 85% wore different footwear between sessions and 75% wore a different *type* of footwear between sessions. Of those who wore a different type of footwear between sessions, 75% had a notable difference in heel height (Appendix A). When comparing this group to $D_{AS,AF}$ as a whole, the proportions of athletic and flat canvas shoes were 18% and 7% lower, respectively, whereas the proportions of women’s ankle boots (heeled), men’s business shoes, and women’s business shoes were 15%, 14%, and 4% higher, respectively. Finally, the proportion of females in this group was 11% higher (65% female vs. 54% female in $D_{AS,AF}$).

### 4.2. Ablations

Across all models, the eight component input led to the highest mean accuracy on all but $D_{PS,SF}$ (Table 5). Using the eight component input, mean accuracy was higher on subsets with samples from all speed categories, compared to subsets with preferred speed samples only. Conversely, using an individual component as the input, mean accuracy was lower on subsets with samples from all speed categories compared to subsets with preferred speed samples only. These relationships held when examining each architecture independently, except in FCNN models (where, for a given input, mean accuracy was nearly always higher on subsets with samples from all speed categories compared to subsets with samples from preferred speed only) and CNN models (where no clear relationship was observed).

## 5. Discussion

In this work, DML was applied for zero-shot re-ID using a large force platform dataset that was purpose built for evaluating person re-ID systems and, thus, had the most complex set of walking conditions. Several baseline DNN architectures of varying types and complexities were implemented on four different data subsets to provide scope for future DNN design and to determine the effects of walking speed and footwear on re-ID performance. The results should be interpreted in the context of the multiple challenges introduced in the evaluation protocol. Namely, re-ID performance was evaluated in the zero-shot setting, with the additional constraints that each query sample had to be from session two and could only be compared with one random prior sample per ID from session one to generate a prediction. To illustrate the effect of constraining the number of priors, mean test accuracy of the FCNN models over seven-fold cross-validation (six sets of 29 IDs and one set of 19 IDs) was 85% (Table 3) with the constraint applied, yet 94% without the constraint applied (Appendix A ‘All’). As this constraint was not applied in previous studies, the value of 94% (referred to herein as the present benchmark) was used to draw comparisons.

When compared to other studies on force platform-based re-ID, the present benchmark was considerably higher than the previous benchmark at many-shot re-ID conducted across measurement sessions (86% on 79 IDs), yet lower than the previous benchmarks for many-shot re-ID conducted within a single session (99%+) [18,19,20,39]. When compared to vision-based re-ID systems, the present benchmark was notably higher than the accuracy of the current state-of-the-art system on CASIA-B under the most comparable conditions (82%) [27]. Namely, this accuracy was obtained on a similar number of IDs (50) in the condition where clothing differed between measurement instances via the addition of a coat. This provides early evidence that force platform-based re-ID systems could be competitive with video-based re-ID systems. However, caution should be exercised when comparing results between this study and others—particularly those in vision-based re-ID—because there are numerous potential sources of variation, other than those discussed here (e.g., different datasets, input representations, model designs, training and evaluation protocols, and implementation details).

The first main finding of this work was that models with the simplest architecture (i.e., FCNN models) performed the best on all four subsets. All models achieved above 97% accuracy during training, suggesting a similar level of fit in training sets. However, FCNN models required more epochs (on average) to reach this level of accuracy compared to others (approximate difference: 26–37 on $D_{AS,AF}$, 28–34 on $D_{AS,SF}$, 11–31 on $D_{PS,AF}$, and 9–33 on $D_{PS,SF}$). The fact that FCNN models learnt more slowly than other models (particularly on subsets with more samples) may suggest the discovery of more generic features. Thus, the utility of more complex DNN architectures, such as those implemented in recent works, is called into question when working with datasets containing just a few thousand samples (even when such datasets contain intra- and inter-individual variations in clothing, walking speed, and footwear, as in $D_{AS,AF}$) [18,19,20].

The second main finding of this work was that footwear was the main impediment to accuracy across all models on $D_{AS,AF}$. A preliminary investigation into the impact of footwear by Derlatka suggested that changes in footwear between measurement instances complicates force platform-based re-ID using a classification framework [24]. The present results suggest that cross-footwear comparisons also complicate re-ID using DML models on much larger datasets (114–193 IDs, compared to 40). The degree of difficulty appears to scale according to the amount of difference in footwear. This was evidenced by the fact that 85% of those who were most commonly mistaken wore different footwear between sessions, and 75% wore different *types* of footwear between sessions. Changes in heel height and sole stiffness, in particular, may pose a challenge, based on the prevalence of changes in heel height and the increased proportion of (presumably) hard-soled footwear (e.g., ankle boots and business shoes) compared to soft-soled footwear (e.g., athletic shoes) in the commonly mistaken group. In 2017, Derlatka compared athletic shoes to high heels and found that the accuracy of a kNN classifier only dropped considerably when high heels were encountered during testing but not training [17]. Thus, the problem posed by footwear type may have been two-fold: shoes that were less represented in the dataset also had characteristics, such as heels, that appear to have complicated re-ID.

The breakdown of accuracy by walking speed revealed why accuracy was slightly higher on $D_{AS,AF}$, compared to $D_{PS,AF}$. Including samples from faster and slower than preferred speed introduced cross-speed comparisons, which had a negative effect on overall accuracy. However, this was offset by a 6.1% increase in accuracy on preferred–preferred speed comparisons, as well as the inclusion of fast–fast and slow–slow speed comparisons, which were relatively easy. In line with this, Moustakidis, Theocharis, and Giakas found negligible change in classification accuracy on a test set of preferred speed samples when their training set contained samples from all speed categories, compared to just preferred speed samples [16]. These findings suggest that increasing the number of training samples (distributed largely evenly across IDs) can augment test performance, even if the test samples are from a different speed category.

Ablation studies revealed that the eight component input was generally only best on subsets with all speed categories (i.e., $D_{AS,AF}$ and $D_{AS,SF}$). Perhaps this is because the additional information encoded in eight components (compared to one) was only necessary when task complexity increased due to the introduction of cross-speed comparisons. Another possible explanation is that the additional information could only be extracted effectively when the number of samples reached a certain threshold somewhere between 1900 (as for $D_{PS,AF}$) and 3298 (as for $D_{AS,SF}$). In the case of FCNN models, the fact that the eight component input was best on all four subsets suggests that the single channel architecture contained too few parameters to extract meaningful information. As such, the optimal number of parameters to fit the current dataset for zero-shot re-ID could be between that of the single channel FCNN (226,050) and the more complex architectures (≈15,305,000). Up until now, force platform-based re-ID systems have included just a few select components of force platform data (either in their entirety or a reduced set of features thereof), based on their presumed utility for the task [6]. The results from this study suggest that it is more effective to include all components when implementing DNNs on relatively large and complex datasets.

This study was not without limitations. In terms of demographics and experimental protocol, individuals with gait disorders were excluded, despite representing approximately 30% of the elderly population (60+ years) and presumably a smaller, yet considerable, proportion of the adult population [56]. Data were acquired over a short-term of 3–14 days. Certain applications may require longer term re-ID which has been shown to be more difficult using both video and force platform data [24,57]. Furthermore, object carriage conditions were not included, yet objects alter effective body mass and the location of the center of mass. This causes adaptations in gait that could complicate the re-ID task [58,59]. These conditions simplified the experimental protocol and facilitated investigation into the baseline utility of DML models for force platform-based re-ID. In terms of the dataset, walking trials were only included if each foot landed completely within the sensing area of each force platform. Enforcing this task constraint would be inconvenient in authentication applications because users would likely have to alter stride timing which might alter force patterns [60]. It would be impossible in surveillance applications, because the goal is to have subjects unaware of the system. Finally, in $D_{AS,AF}$ and $D_{AS,SF}$, samples were distributed approximately equally among speed categories to facilitate fair comparison; however, in practice, it would be relatively rare for an individual to walk significantly slower or faster than their preferred speed.

Future work should investigate strategies to improve performance on cross-speed and cross-footwear comparisons without compromising performance elsewhere. Based on the findings from this study, a promising avenue to explore is increasing training set size. It would be valuable to determine the independent effects of training set size (i.e., total number of samples) and sample distribution (between IDs or conditions such as walking speeds). With respect to sample distribution, one recommendation is to include slower and faster than preferred speed samples sparingly to better represent the speed distribution expected in most applications. Other avenues to explore include the following: time-frequency representations; alternative loss functions; more refined DNN architectures; and more advanced training methods (e.g., transfer learning and data augmentation). The independent and combined evaluation of these strategies would provide a clearer picture of the upper limit of performance. On top of this, future work should quantify the effects of gait disorders, object carriage, and partial foot contacts on both short and long-term re-ID, particularly in the challenging zero-shot setting.

## 6. Conclusions

Force platforms are promising sensors for gait-based person re-ID because they can be applied in settings with limited or no visibility and are a rich source of kinetic information. The purpose of this work was to evaluate several baseline DNN architectures for zero-shot re-ID using a DML framework. This was done on four different subsets of a large, purpose built force platform dataset that contained significant variations in task and environmental constraints. Two-layer FCNN models slightly outperformed models with more complex architectures, achieving 85% accuracy on the entire dataset in a challenging evaluation setting. A breakdown of performance across all models revealed that re-ID accuracy declined according to the amount of change in individuals’ walking speed and footwear between measurement instances. Overall, this work confirmed the baseline utility of DML models for scalable gait-based person re-ID using force platform data.

## Figures and Tables

**Figure 1 sensors-23-03392-f001:**
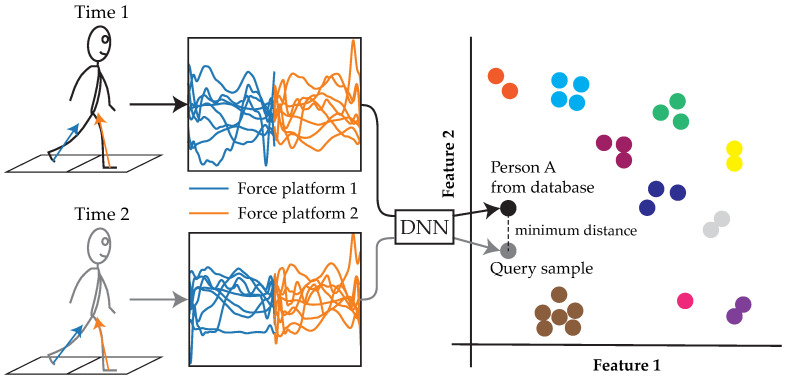
Workflow of person re-identification (re-ID) using a using a deep metric learning model. An unknown identity (ID) can be re-identified by extracting their gait features using a deep neural network (DNN) and comparing them to those of other samples in the database using a distance metric. Following this approach, the same trained model can be used to re-identify any number of IDs who were not seen during training. Unlabeled colors each represent a different ID. The feature space is depicted as two-dimensional for simplicity.

**Figure 2 sensors-23-03392-f002:**
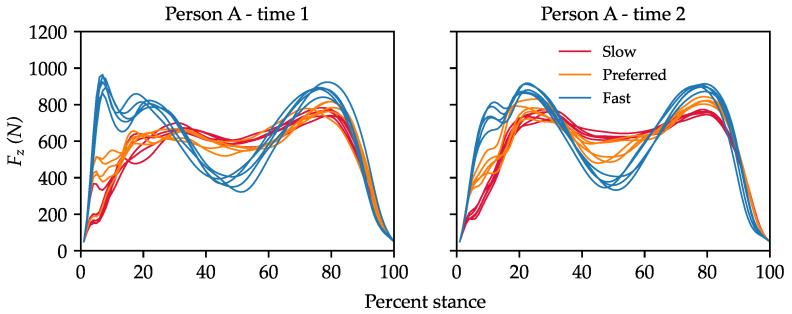
An example of the vertical ground reaction force ($F_{z}$) with changes in walking speed (colors) and footwear (**left** vs. **right**) between measurement instances. **Left**—women’s ankle boots (flat sole); **right**—ballet flats.

**Figure 3 sensors-23-03392-f003:**
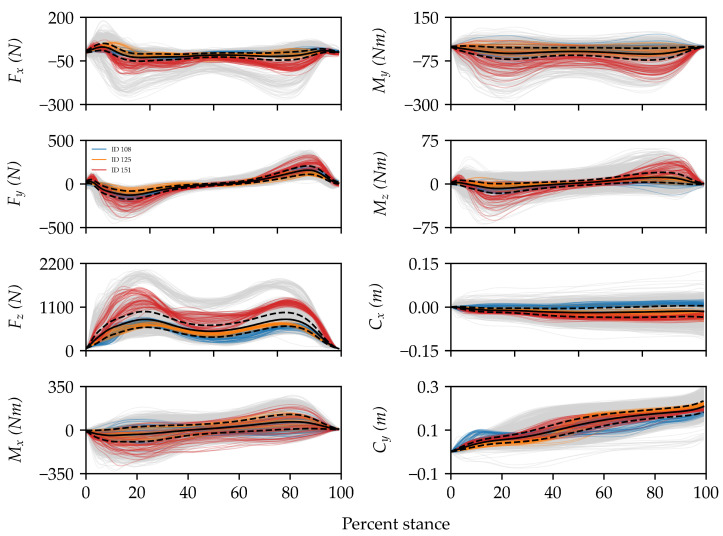
Visualization of all ground reaction forces ($F_{x}$—mediolateral, $F_{y}$—anteroposterior, and $F_{z}$—vertical), ground reaction moments ($M_{x}$, $M_{y}$, and $M_{z}$), and center of pressure coordinates ($C_{x}$ and $C_{y}$) for three random IDs in the dataset (colors) compared to the rest (grey). Solid and dashed black lines are the means and standard deviations across the entire dataset, respectively. For ease of interpretation, the signals have been clipped, filtered, and time normalized to 100% of the stance phase. Overall, there is a large degree of variability in the dataset, and some components are more variable than others.

**Figure 4 sensors-23-03392-f004:**
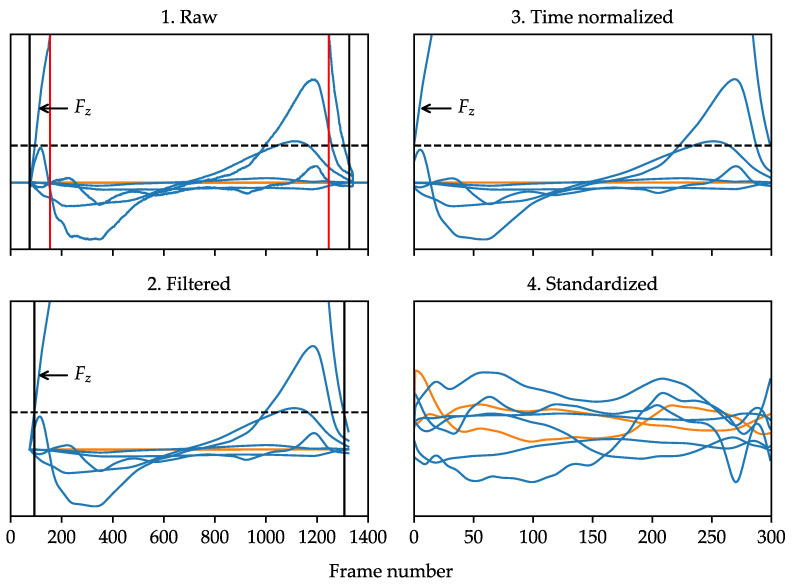
Illustration of the signal pre-processing method for ground reaction forces (GRFs) and ground reaction moments (GRMs) (blue) compared to center of pressure (COP) coordinates (orange). Vertical black lines indicate clip points for GRFs and GRMs. Vertical red lines indicate clip points for COP coordinates. The horizontal dashed line is at 50 N magnitude. The Y-axis limits excluded most of $F_{z}$ so that other components were easier to see.

**Figure 5 sensors-23-03392-f005:**
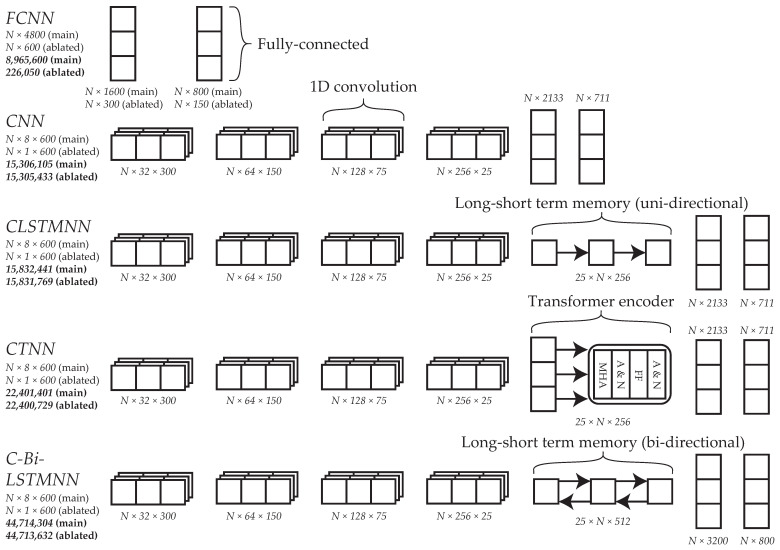
Overview of the DNN architectures used in this study. Shown on the left is the name of each architecture, the shape of its input, and its total number of parameters (top-to-bottom, respectively). The shape notation is number of samples N× number of features *F* for the fully connected neural network (FCNN) and number of samples N× number of channels C× sequence length *L* for the others. CNN = convolutional neural network, CLSTMNN = convolutional long short-term memory neural network, CTNN = convolutional transformer neural network, C-Bi-LSTMNN = convolutional bi-directional long short-term memory neural network.

**Table 1 sensors-23-03392-t001:** Overview of key characteristics of public force platform datasets containing healthy individuals. Gutenberg gait database comprises 10 datasets, each used for a separate study. All sessions were conducted on the same day in GaitRec (healthy). * Clothing conditions were not mentioned for datasets other than ForceID A and, thus, were inferred where possible. S = slower than preferred, P = preferred, F = faster than preferred.

Dataset	No. IDs	No. Sessions	Footwear	Walking Speed	Object Carriage	Clothing *
AIST gait database [43]	300	1	Barefoot	P	-	Skin-tight
Gutenberg gait database [41]	350	1 (253 IDs) 2 (47 IDs) 6 (42 IDs) 8 (8 IDs)	Barefoot	P	-	Skin-tight
GaitRec (healthy) [44]	211	1–6	Barefoot and personal	S, P, F	-	Skin-tight
Derlatka & Borowska (2023) [42]	324	1 (294 IDs) 2 (30 IDs)	Personal (semi-controlled)	P	-	Unknown
ForceID A (ours)	184	2 (184 IDs)	Personal	S, P, F	-	Personal

**Table 2 sensors-23-03392-t002:** Distribution of footwear in the dataset.

Footwear	Session One Count	Session Two Count	Total
Athletic	80	76	156
Flat canvas (slip-on/laced)	45	49	94
Women’s ankle boot (flat)	11	14	25
Ballet flat	11	14	25
Men’s business	13	11	24
Women’s ankle boot (heel)	10	9	19
Men’s ankle boot	4	5	9
Sandal	5	4	9
Flip-flop	4	5	9
Loafer	4	4	8
Women’s business	2	0	2
Steel capped boot	1	1	2
Five finger	1	1	2
Rubber boot	1	0	1
Unknown	1	0	1

**Table 3 sensors-23-03392-t003:** Accuracy (A) and F1 score (F) of each deep neural network architecture on each data subset (*D*) over seven-fold cross-validation (provided as mean and min–max range over test sets with the best results in bold). The results for the fully connected neural network (FCNN) architecture on ForceID A are also given as a benchmark. CNN = convolutional neural network, CLSTMNN = convolutional long short-term memeory neural network, CTNN = convolutional transformer neural network, C-Bi-LSTMNN = convolutional bi-directional long short-term memory neural network, AS = all speeds, PS = preferred speed, AF = all footwear, SF = same footwear (between sessions).

Subset	Architecture	Test Performance	
		A (%)	F
$D_{AS,AF}$	FCNN	85.42(80.76 **–** 89.54)	0.85(0.80 **–** 0.89)
	CNN	83.12(79.66–87.92)	0.83(0.79–0.87)
	CLSTMNN	83.75(79.18–87.45)	0.83(0.78–0.87)
	CTNN	85.31(79.69–87.97)	0.85(0.79–0.88)
	C-Bi-LSTMNN	85.33(80.31–88.74)	0.85(0.79–0.88)
$D_{AS,SF}$	FCNN	91.88(85.26 **–** 97.18)	0.92(0.85 **–** 0.97)
	CNN	89.23(81.92–96.32)	0.89(0.82–0.96)
	CLSTMNN	89.71(82.64–96.45)	0.90(0.82–0.96)
	CTNN	91.10(83.59–97.88)	0.91(0.83–0.98)
	C-Bi-LSTMNN	89.82(81.84–96.36)	0.90(0.82–0.96)
$D_{PS,AF}$	FCNN	86.77(81.68 **–** 90.42)	0.86(0.80 **–** 0.90)
	CNN	80.98(76.94–87.74)	0.79(0.74–0.88)
	CLSTMNN	82.34(77.63–88.51)	0.81(0.76–0.88)
	CTNN	82.84(78.02–88.89)	0.82(0.76–0.89)
	C-Bi-LSTMNN	85.12(78.66–89.66)	0.84(0.77–0.89)
$D_{PS,SF}$	FCNN	91.74(86.03 **–** 97.43)	0.91(0.85 **–** 0.97)
	CNN	88.91(79.04–95.96)	0.88(0.78–0.96)
	CLSTMNN	88.65(80.88–95.96)	0.88(0.80–0.96)
	CTNN	89.99(77.21–95.59)	0.90(0.77–0.96)
	C-Bi-LSTMNN	87.59(79.78–96.32)	0.87(0.79–0.96)
Force ID A	FCNN	85.57(81.04–90.90)	0.85(0.81–0.91)

**Table 4 sensors-23-03392-t004:** Accuracy (A) across all test sets within $D_{AS,AF}$ for specific between-session speed and footwear comparisons. The highest A is in bold.

Footwear	Walking Speed	Walking Speed Comparison	No. Predictions	A (%)
Same	Same	F—F	6760	**95.72**
		S—S	6985	95.26
		P—P	7205	94.13
	Cross	S—P	14,100	92.05
		P—F	13,600	89.49
		S—F	13,600	81.33
Cross	Same	F—F	4345	83.22
		S—S	4830	82.92
		P—P	4940	82.15
	Cross	S—P	9795	78.25
		P—F	9275	71.54
		S—F	9295	68.33
All			104,730	84.45

**Table 5 sensors-23-03392-t005:** Accuracy (A) across all models with different inputs (provided as mean and min–max range over test sets with the best results in bold).

Input Component/s	A (%)			
	$D_{\mathrm{AS},\mathrm{AF}}$	$D_{\mathrm{AS},\mathrm{SF}}$	$D_{\mathrm{PS},\mathrm{AF}}$	$D_{\mathrm{PS},\mathrm{SF}}$
$C_{x}$	32.03(20.96–46.25)	38.27(22.80–52.21)	38.22(26.29–54.79)	41.86(27.21–57.35)
$M_{z}$	33.54(22.05–44.67)	38.73(26.00–50.75)	39.67(29.31–51.29)	42.26(21.88–61.40)
$M_{y}$	38.39(25.22–56.63)	41.91(20.62–60.67)	39.22(19.40–53.62)	37.30(19.85–56.25)
$M_{x}$	48.27(36.82–60.03)	56.74(43.18–68.85)	51.55(29.53–70.39)	48.47(25.37–74.48)
$C_{y}$	46.94(34.89–58.25)	65.66(47.63–75.32)	54.93(47.20–64.44)	72.88(59.38–87.13)
$F_{x}$	53.90(35.78–68.29)	60.81(44.57–75.95)	65.72(45.69–77.30)	61.78(42.28–85.94)
$F_{y}$	56.12(38.42–65.52)	70.41(55.29–77.78)	69.29(55.56–81.91)	81.26(63.60–91.15)
$F_{z}$	76.62(66.27–86.53)	85.75(73.86–93.18)	83.11(72.63–94.41)	91.72(77.21 **–** 99.76)
All	84.59(79.18 **–** 89.54)	90.35(81.84 **–** 97.88)	83.61(76.94 **–** 90.42)	89.38(77.21–97.43)

## Data Availability

The data presented in this study are openly available in figshare at https://doi.org/10.25909/14482980.v6. Accessed on 29 September 2022.

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
