# Peer review of "Deep Metric Learning for Scalable Gait-Based Person Re-Identification Using Force Platform Data"

_sensors, 2023, doi:10.3390/s23073392_

Round 1

Reviewer 1 Report

Although the manuscript is interesting and written in a good English, there are several points where further improvements and explanations are required from the authors. Although person re-identification is a hot research topic both in academia and industry, the authors did not write a related work section. Specifically, the authors did not put their work into context. First, the authors should cite the different approaches of person re-identification. For example, one can find computer vision approaches (Person Re-identification based on Deep Multi-instance Learning, 2017) where similar loss functions were used. More importantly, the architecture of the proposed network remains somewhat unclear. Besides Figure 5, some more detailed explanations are required since the exact architecture is rather unclear. Could you compare the proposed method to at least one other state-of-the-art method?

Reviewer 2 Report

This paper presents a new gait dataset that contains the gait sequences of 193 subjects. The dataset is collected using force platforms which measure ground reaction forces along three dimensions. The authors also evaluated different deep learning models on their collected dataset for person re-identification tasks.

To me, the major contribution of the manuscript is the proposal of a new gait dataset. I would suggest improving the description of the dataset in the readme file so that the research community can also get the benefit. Furthermore, the authors should highlight any new challenges to the research community as compared to the existing ones. For a detailed comparison of different datasets, I would like to suggest the following recent article:

Khan, Muhammad Hassan, Muhammad Shahid Farid, and Marcin Grzegorzek. "Vision-bsed approaches towards person identification using gait." Computer Science Review 42 (2021): 100432.

A few datasets are available too in this domain of research; the authors need to provide some indications and analysis on how the presented gait dataset is challenging and useful to the research community in terms of covariates.

The authors shall explain the reason why low-pass filtering and normalization techniques are applied to raw data.

The process of the construction of the gait-cycle shall be explained precisely.

How input (to deep networks) are constructed? The authors shall elaborate this process for each of the networks in the description.

A few abbreviations are not defined clearly in the manuscript e.g., Section 2.3

I would suggest improving the description of the code in the readme file so that the research community can also get the benefit.

Round 2

Reviewer 1 Report

I think the manuscript can be accepted now. The authors improved the manuscript significantly by answering my questions.

Reviewer 2 Report

Since the authors have addressed all the points, I would like to accept this article.